# Safety and Neuroprotective Efficacy of Palm Oil and Tocotrienol-Rich Fraction from Palm Oil: A Systematic Review

**DOI:** 10.3390/nu12020521

**Published:** 2020-02-18

**Authors:** Maznah Ismail, Abdulsamad Alsalahi, Mustapha Umar Imam, Der Jiun Ooi, Huzwah Khaza’ai, Musheer A. Aljaberi, Mad Nasir Shamsudin, Zulkifli Idrus

**Affiliations:** 1Laboratory of Molecular Biomedicine, Institute of Bioscience, Universiti Putra Malaysia, Serdang 43400 UPM, Selangor, Malaysia; ahmedsamad28@yahoo.com; 2Department of Medical Biochemistry, Faculty of Basic Medical Sciences, Usmanu Danfodiyo University, 840231 Sokoto, Nigeria; mustyimam@gmail.com; 3Centre for Advanced Medical Research and Training, Usmanu Danfodiyo University, 840231 Sokoto, Nigeria; 4Department of Oral Biology & Biomedical Sciences, Faculty of Dentistry, MAHSA University, Jenjarom 42610, Selangor, Malaysia; djooi@mahsa.edu.my; 5Department of Biomedical Sciences, Faculty of Medicine and Health Sciences, Universiti Putra Malaysia, Serdang 43400 UPM, Selangor, Malaysia; huzwah@upm.edu.my; 6Community Health Department, Faculty of Medicine and Health Sciences, Universiti Putra Malaysia, Serdang 43400 UPM, Selangor, Malaysia; gabrisyria@yahoo.com; 7Department of Agribusiness and Bioresource Economics, Faculty of Agriculture, Universiti Putra Malaysia, Serdang 43400 UPM, Selangor, Malaysia; mns@upm.edu.my; 8Institute of Tropical Agriculture and Food Security, Universiti Putra Malaysia, Serdang 43400 UPM, Selangor, Malaysia; zulidrus@upm.edu.my

**Keywords:** cognition, neurodegeneration, neuroprotection, palm oil, tocotrienol-rich fraction, tocotrienol

## Abstract

Background: Several natural products have been reported to elicit beneficial effects against neurodegenerative disorders due to their vitamin E contents. However, the neuroprotective efficacy of palm oil or its tocotrienol-rich fraction (TRF) from the pre-clinical cell and animal studies have not been systematically reviewed. Methods: The protocol for this systematic review was registered in “PROSPERO” (CRD42019150408). This review followed the Preferred Reporting Items for Systematic Reviews and Meta-Analysis (PRISMA) guidelines. The Medical Subject Heading (MeSH) descriptors of PubMed with Boolean operators were used to construct keywords, including (“Palm Oil”[Mesh]) AND “Nervous System”[Mesh], (“Palm Oil”[Mesh]) AND “Neurodegenerative Diseases”[Mesh], (“Palm Oil”[Mesh]) AND “Brain”[Mesh], and (“Palm Oil”[Mesh]) AND “Cognition”[Mesh], to retrieve the pertinent records from PubMed, Scopus, Web of Science and ScienceDirect from 1990 to 2019, while bibliographies, ProQuest and Google Scholar were searched to ensure a comprehensive identification of relevant articles. Two independent investigators were involved at every stage of the systematic review, while discrepancies were resolved through discussion with a third investigator. Results: All of the 18 included studies in this review (10 animal and eight cell studies) showed that palm oil and TRF enhanced the cognitive performance of healthy animals. In diabetes-induced rats, TRF and α-tocotrienol enhanced cognitive function and exerted antioxidant, anti-apoptotic and anti-inflammatory activities, while in a transgenic Alzheimer’s disease (AD) animal model, TRF enhanced the cognitive function and reduced the deposition of β-amyloid by altering the expression of several genes related to AD and neuroprotection. In cell studies, simultaneous treatment with α-tocotrienols and neurotoxins improved the redox status in neuronal cells better than γ- and δ-tocotrienols. Both pre-treatment and post-treatment with α-tocotrienol relative to oxidative insults were able to enhance the survival of neuronal cells via increased antioxidant responses. Conclusions: Palm oil and its TRF enhanced the cognitive functions of healthy animals, while TRF and α-tocotrienol enhanced the cognitive performance with attenuation of oxidative stress, neuroinflammation and apoptosis in diabetes-induced or transgenic AD animal models. In cell studies, TRF and α-tocotrienol exerted prophylactic neuroprotective effects, while α-tocotrienol exerted therapeutic neuroprotective effects that were superior to those of γ- and δ-tocotrienol isomers.

## 1. Introduction

Neurodegeneration covers a wide range of pathological processes, and is characterized by progressive neuronal loss, leading to impaired neuronal function and death [1]. Several neurodegenerative disorders begin with the deposition of noxious misfolded proteins, such as β-amyloid and tau proteins in Alzheimer’s disease (AD) or α-synuclein in Parkinson’s disease (PD), on neuronal cells [2,3].

AD involves neurodegeneration in the hippocampal and entorhinal cortical neurons, while in PD, dopaminergic neurons of the substantia nigra are affected [4]. Several cerebral events of interrelated pathways are involved in the etiopathogenesis of neurodegeneration, including altered gene expression, neuroinflammation, neurotoxicity, and oxidative stress, which eventually result in neuronal cell death [5]. The clinical manifestations of these effects include sensory, cognitive or locomotive dysfunction [6].

Although there are available medicines used for neurodegenerative disorders, there is the need for more therapies because these medicine do not remedy the whole spectrum of pathobiology found in neurodegenerative processes [7]. Moreover, the prevalence of neurodegenerative disorders is increasing worldwide [8], and as progressive diseases [1], damaged neurons do not regenerate by themselves [4], further highlighting the need for more effective therapies. Effective therapies for these diseases can be achieved not only by the prophylactic prevention of degenerations but also by delaying their progression once it is initiated [8,9]. Although most therapies for neurodegenerative disorders do not directly alter the underlying etiology of the diseases, the interventions could still offer relief through retarding their development and progression [7].

Phytochemicals from fruits, vegetables, nuts and herbs are reported to exert neuroprotective effects against such diseases as AD and PD, mediated via altering neurogenerative processes related to misfolded protein formation and neuroinflammation [8,10]. Neurodegenerative disorders share a common etiological mechanism of aggregation of noxious misfolded proteins [2,3], suggesting that a parallel strategy of treatment to prevent the disposition of such proteins could be effective across several of these disorders [2].

Palm oil is an edible oil that is extracted from the palm plant fruit (*Elaeis guineensis*) and is one of the most produced and highest consumed edible oils worldwide. This is despite being harvested from the smallest global percentage of all the cultivated land for oils and fats [11]. For such reasons, the phytochemistry of palm oil fruit has been extensively studied and is well established as comprising several bioactive compounds, including a palmitic-oleic-rich semi-solid fat (the main bulk of the oil, with 95% triglycerides and fatty acids in the form of palmitic, myristic, stearic, oleic and linoleic acids), vitamin E fraction (30% tocopherols and 70% tocotrienols), carotenoids, polyphenols and phytosterols [12]. The literature is rich in terms of studies that report the dietary and pharmacological effects of palm oil and its bioactives, including those related to neuroprotection. In this regard, the neuroprotective effects of the tocopherols have been more extensively studied than those of the tocotrienols [13]. Interestingly, several reviews have outlined the pharmacological properties of palm oil and its bioactives in several health conditions [11,14,15,16,17,18]. In this regard, several preclinical and clinical studies have been conducted on the neuroprotective effects of palm oil and its bioactives, although these have not been systematically reviewed anywhere, to the best of our knowledge. Thus, we wanted to review the literature to establish whether there is enough pre-clinical evidence for the neuroprotective effects of palm oil and its bioactives.

## 2. Materials and Methods

The design of the current systematic review followed the Preferred Reporting Items for Systematic Reviews and Meta-Analysis (PRISMA) [19,20]. All the retrieved primary records were preclinical cell (in vitro) and animal (in vivo) studies related to neuroprotective properties of palm oil and its bioactives in exposed cells and animals. Two independent investigators were involved at every stage of the systematic review (retrieval of records from databases, selection of primary and secondary records, risk assessment for bias and data extraction). Throughout, discrepancies were resolved through discussion with a third investigator. The protocol for this systematic review was registered in the international prospective register of systematic reviews database “PROSPERO” (CRD42019150408) (https://www.crd.york.ac.uk/prospero/display_record.php?RecordID=150408). Since this systematic review relied on published preclinical studies and no human studies were included, the informed consent of patients or the approval of the Institutional Review Board (IRB) was not required.

### 2.1. Sereach Methodology

The Medical Subject Heading (MeSH) descriptors of PubMed with Boolean operators were used to construct keywords, including (“Palm Oil”[Mesh]) AND “Nervous System”[Mesh], (“Palm Oil”[Mesh]) AND “Neurodegenerative Diseases”[Mesh], (“Palm Oil”[Mesh]) AND “Brain”[Mesh], and (“Palm Oil”[Mesh]) AND “Cognition”[Mesh]. The pertinent records were retrieved from PubMed, Scopus, Web of Science and ScienceDirect from 1990 to 2019 without refining languages, countries and types of articles. In addition, bibliographies were searched for relevant records, while ProQuest and Google scholar were searched to ensure the comprehensive identification of relevant articles.

### 2.2. Records Identification

The total number of the identified records from the retrieved databases and extra sources were recorded, and the number of results per hit in each database was recorded. Then, duplicated records were removed.

### 2.3. Primary Selection

To restrict selection of the retrieved records to research articles, the records were screened by titles, abstracts or full texts to remove books, reviews, conference abstracts and miscellaneous publications (e.g., indices, glossaries, lists and bibliographies). Then, the identified primary research articles were further screened for relevant content by titles, abstracts or full texts, after which irrelevant primary research articles were excluded.

### 2.4. Secondary Selection

Relevant primary research articles were screened for relevant content by full text according to the prespecified inclusion and exclusion criteria of eligibility. Then, the relevant secondary records were categorized into preclinical cell (in vitro) and animal (in vivo) studies.

#### Eligibility Criteria

The inclusion and exclusion criteria for preclinical cell and animal studies are summarized in Table 1 and Table 2, respectively.

### 2.5. Assessment of the Risks of Bias

The preclinical studies included in this review were assessed for risks of bias to ensure that the outcomes of the studies were not affected by the prevailing bias in the primary studies.

#### 2.5.1. Preclinical Cell Studies

The oral health assessment tool-validity and reliability (OHAT) tool for the assessment of bias in in vitro studies [21] was used to categorize the studies into DL (Definitely Low risk of bias indicates direct evidence of low risk-of-bias practices), PL (Probably Low risk of bias indicates indirect evidence of low risk-of-bias practices or it is deemed that deviations from low risk-of-bias practices during the study would not appreciably bias the results), PH (Probably High risk of bias indicates indirect evidence of high risk-of-bias practices or there is insufficient information, also abbreviated as NR) or DH (Definitely High risk of bias indicates direct evidence of high risk-of-bias practices).

For selection bias, randomization was answered as PL in all studies, since it was deemed that the absence of randomization was unlikely to bias the results. Since allocation concealment is not reported in most cell studies, the answer was NR (insufficient information indicating PH).

For performance bias (during treatment), blinding and the use of an identical vehicle for the intervention and control groups were considered. Therefore, answers were taken as DL (identical type and volume of the vehicle), PL (identical type of vehicle but volume was not reported), NR (insufficient information indicating PH) or DH (use of different vehicle). For blinding, the answer was either DL (if blinding was reported), PL (if indirect evidence that blinding was undertaken), NR (insufficient information indicating PH) or DH (if blinding was reported to be not undertaken).

For detection bias, the answer for the accuracy of exposure characterization was either DL (if the purity of the intervention substance was reported), PL (if the purity of the intervention substance was not reported, but the supplier was reported), NR (insufficient information indicating PH) or DH (if it was reported that a low-grade or impure intervention substance was used), while the answer for consistent exposure was either DL (quantities and timing of exposure were consistent), PL (indirect evidence that the quantities and timing of exposure were consistent), NR (insufficient information indicating PH) or DH (different quantities and timing of exposure was reported). For blinding, the answer was DL (if blinding was reported), PL (indirect evidence that blinding was undertaken), NR (insufficient information indicating PH) or DH (if blinding was reported to be not undertaken).

For attrition bias, the answer for incomplete outcome data was either DL (reported and adequately addressed), PL (indirect evidence that attrition was adequately addressed), NR (insufficient information indicating PH) or DH (attrition was reported but inadequately addressed).

For reporting bias, the answer was either DL (all the primary and secondary outcomes are reported as compared to methods and abstract), PL (indirect evidence of reporting all the primary and secondary outcomes as compared to the methods and abstract), PH (insufficient information indicating PH) or DH (some of the primary outcomes were not reported as compared to the methods and abstract).

For other sources of bias, the answer was either DL (appropriate statistics and/or adherence to the protocol when comparing the reported outcomes to objectives and methods), PL (indirect evidence of appropriate statistics and/or adherence to the protocol when comparing the reported outcomes to objectives and methods), or PH (direct evidence of inappropriate statistics and/or non-adherence to the protocol when comparing the reported outcomes to objectives and methods). Finally, if more than four domains are recorded to be PH, the study was excluded, because there is probably a high risk of bias.

#### 2.5.2. Preclinical Animal Studies

The assessment of the risk of bias in animal studies was performed according to the SYRCLE’s tool [22]. The answer for the signaling questions was either “Yes” to indicate a low risk of bias, “No” to indicate a high risk of bias or “U” to indicate an uncertain level of bias.

For the assessment of selection bias (before starting intervention), the baseline characteristics were adopted in place of allocation concealment, since baseline characteristics are standard items in the assessment of selection bias and randomization is not a standard practice in animal studies [22]. However, randomization was considered since it is reported in most animal studies. In response to the research question of this systematic review, the baseline characteristics adopted were a similar age, sex, species and strain of animals, similar timing and characteristics of intervention, similar health status of animals (healthy or disease-induced), similar genetic background of animals (normal or transgenic), similar source of animals, similar inducer (type and dose), conditions and procedures of disease induction (in case of chemical induction of a disease), similar conditions, procedure and period of surgery and recovery (in cases of surgical induction of a disease), isocaloric diet, identical vehicle for either pre-, during or post-gestation in dams–offspring studies. Accordingly, the answer for randomization was Yes (if randomization was explicitly reported) or No (if randomization was not reported), while the answer for the baseline characteristics was either Yes (if similar baseline characteristics were reported) or No (if the baseline characteristics were different).

For the assessment of performance bias (during intervention), randomization was answered as either Yes (if a random housing of cages was reported), or No (if randomization of cages during housing was not reported). For blinding, the answer was Yes (if blinding was reported or it is deemed to have been performed by another person) or No (if blinding was not reported or it was evidenced that the same person performed the treatment, measured the outcomes and analyzed the outcome).

For detection bias (During the outcome measurement), the answer was Yes (if randomized selection of animals was reported for measuring the outcomes and/or gathering blood sample, biopsy or tissues) or No (if the random selection was not reported). For blinded detection, the answer was either Yes (if blinding was reported or there was evidence that another person performed the outcome measurement) or No (if blinding was not reported or there was evidence that the same person performed and analyzed the outcome measurement).

For attrition bias (exclusion of animals during the treatment period because of death, disability, infection or injury/exclusion of data as outliers during the analysis of the outcomes), the answer was either Yes (if the authors reported and addressed the incomplete outcome data adequately, or the incomplete outcome data were not reported but there was evidence of balanced sample size from the time of randomization until outcome reporting) or No (if adequate addressing of incomplete data was not reported or there was evidence that the author did not report imbalanced sample size from the time of randomization to outcome reporting).

For selective outcome reporting, this systematic review adjusted the query for comparing the reported outcomes with methods because the protocols for animal studies are not always available or registered like clinical trials [22]. In addition, the query was extended to assess whether authors reported all the pre-specified primary and secondary outcomes or selectively reported the primary and secondary outcomes. Accordingly, the answer was either Yes (if all the pre-specified primary or secondary outcomes were reported using the same pre-specified measurements for the primary and secondary outcomes under their own specified methods) or No (if the authors did not report all the pre-specified primary and secondary outcomes or did not report a key primary outcome or reported a primary outcome that was not pre-specified).

For other sources of bias, this systematic review adopted this domain to assess funder bias. If such problems could be inferred directly or indirectly, the answer was No. Otherwise, the answer was U to indicate an uncertain level of bias. Finally, if more than four domains were answered “No”, this indicated a high risk of bias and the study was excluded.

### 2.6. Data Extraction

#### 2.6.1. Preclinical Cell Studies

For study design, the type of study (preclinical in vitro study); challenging characteristics (type of neurotoxin and duration of challenging cells with the neurotoxin); a mono-level intervention group with a negative comparator group and multi-level intervention groups with a comparator group were considered. For human disease models, the type of cells (primary or secondary cells, neuronal slices); source of cells (human, animal or another organism); type of cell line (neuronal or neuroglial); primary cells (source organ, species, sex and strain of animal source, age of the animal at the time of gathering the organ) and genetic background of cells (normal or transgenic) were considered. For intervention, the doses in units; timing of intervention (pre or post or simultaneous with the neurotoxin challenge); duration of the intervention and vehicle of the intervention were used. For primary outcomes, the cytotoxicity (using cellular viability); neuroinflammation (using levels of TNF-α, IL-1β, IL-1α, p65 of NF-κβ); apoptosis (using level of caspase-3); oxidative stress (using levels of glutathione, oxidized glutathione, catalase, glutathione peroxidase and superoxide dismutase for measuring the antioxidant activity, while the level of malondialdehyde was for measuring lipid peroxidation) and neurodegeneration (using levels of misfolded proteins) were used. For secondary outcomes, structural changes in cell morphology (descriptive microscopical abnormalities, immunocytochemistry and immunofluorescence staining); and molecular mechanisms (up- or down-regulation of gene expression and metabolomics) were considered. The extracted data were summarized into a table.

#### 2.6.2. Preclinical Animal Studies

For study design, the type of study (preclinical in vivo animal study); health problem of interest; number of groups (a mono-level intervention group with a comparator group, multi-level intervention groups with a comparator group); human disease model; total number of animals; number of animals per each group; disease induction characteristics (type of induction, type of inducer, dose of inducer, route of induction, duration of induction and recovery period after induction); type of exposure (acute, short-term or long-term) and duration of intervention (hours, days, weeks or months) were considered. For human disease models, the age of animal (pups, young, adult, aged); sex of animal (males of females); species of animal (rats or mice); strain of animal (Sprague–Dawley, Wister, etc.); and genetic background (normal or transgenic) were used. For intervention, the doses in units; timing of intervention; frequencies of dose per day (once or twice, etc.); duration of intervention (hours, days, weeks, months) and route of intervention (ad libitum or calibrated admixed with diet or water, forced oral gavage or parenteral) were used. For primary outcomes, the neurotoxicity (using the level of acetyl cholinesterase); neuroinflammation (using levels of TNF-α, IL-1β, IL-1α and p65 of NF-κβ); apoptosis (using the level of caspase-3); oxidative stress (using levels of reduced glutathione, oxidized glutathione, catalase, glutathione peroxidase and superoxide dismutase for the antioxidant activity, while the level of malondialdehyde for lipid peroxidation); neurodegeneration (levels of misfolded proteins) and cognitive functions (escape latency for measuring spatial learning and memory) were considered. For secondary outcomes, structural changes (descriptive microscopical abnormalities in the histology of organs; morphometric histological abnormalities, immunohistochemistry) and molecular changes (up- or down-regulation of gene expression and metabolomics) were considered. Finally, the extracted data were summarized into a table.

### 2.7. Strategy for Data Synthesis

A qualitative approach was used as the preferred option for this systematic review through a narrative literature synthesis by summarizing the primary and secondary outcomes with a view to systematically present the methodology and findings, and show the limitations, drawbacks and deviations, which could affect the observed outcomes.

## 3. Results

### 3.1. Identified Records

A total of 2076 records were retrieved, out of which 2049 were from the four databases, while 27 were from the other sources (Table 3), and then 654 duplicates were removed (Figure 1).

### 3.2. Primary Selected Records

After the initial screening, there were 1422 articles, from which 895 records were removed, including books (*n* = 437), reviews (*n* = 339) conference abstracts (*n* = 72) and miscellaneous records (*n* = 46). Then, a total of 527 records were identified as research articles, from which 444 unrelated records were removed. Finally, 83 relevant research articles and three theses were selected (Figure 1).

### 3.3. Secondary Selected Records

According to the inclusion and exclusion criteria of eligibility, 43 records were excluded, including modified interventions (*n* = 13), unrelated neuroprotective effects (*n* = 21), an inaccessible article (*n* = 1), in silico studies (*n* = 1) and human studies (*n* = 7). Accordingly, 40 research articles were eligible to be included into the qualitative synthesis of the literature.

### 3.4. Asessment of the Risks of Bias

The assessment of the risk of bias of 40 the eligible records resulted in excluding 22 highly biased records (12 animal studies and 10 cell studies). Finally, the low-risk unbiased records were included into the literature synthesis, comprising 18 studies, namely; eight preclinical cell studies (Table 4) and 10 preclinical animal studies (Table 5).

### 3.5. Outcomes

#### 3.5.1. Preclinical Cell Studies

Cytoprotective effects of tocotrienol-rich fraction (TRF)Simultaneous treatmentTreatment with either 0.1, 1 or 10 µM of TRF at the same time as hydrogen peroxide for 24 h was found to enhance the cellular viability of primary cells of the anterior striatum of foetal Wistar rats (17th–19th day of gestation) [27]. Conversely, treatment with either 0.00003%, 0.0003% or 0.003% of TRF together with Aβ42 aggregates for 24 h did not significantly enhance the cellular viability of a human neuroblastoma cell line (SH-SY5Y) [30], (Table 6).Pre-treatmentA five-minute pre-treatment with 250 nm TRF enhanced the cellular viability of mouse hippocampal HT4 neuronal cells line and cerebrocortical neurons of foetuses of Sprague–Dawley rats (17th day of gestation) when subsequently subjected to a 24-h challenge with glutamate neurotoxin, which exerts its effects through the direct inhibition of inducible 12-lipoxignase enzyme and the inhibition of tyrosine phosphorylation of inducible 12-lipoxignase enzyme [29]. Similarly, a five-minute pre-treatment with 200 ng/mL TRF enhanced the cellular viability and survival, and reduced lipid peroxidation in human neuroblastoma cell line (SK-N-SH) through antinecrotic and antiapoptotic effects (early and late apoptosis) against a 24-h challenge with glutamate neurotoxin [28]. However, a five-minute pre-treatment with 100, 200 or 300 ng/mL of TRF neither enhanced cellular viability nor modulated the redox status in a human astrocytes cell line (CRL-2020 cells) against a 24-h challenge with glutamate neurotoxin. Conversely, 200 and 300 ng/mL of TRF attenuated lipid peroxidation and reduced the percentages of apoptotic and necrotic cells [26], (Table 6).Post-treatmentA 30-min post treatment with 100, 200 and 300 ng/mL of TRF attenuated cellular apoptosis and necrosis, although only 200 ng/mL TRF could significantly recover the cellular viability of a human neuroblastoma cell line (SK-N-SH) challenged with glutamate neurotoxin for 24 h through the maintenance of the cellular membrane integrity. On the other hand, 300 mg/mL of TRF significantly attenuated lipid peroxidation in the same cells [28]. Similarly, a 30-min post-treatment with 100, 200 or 300 ng/mL of TRF neither recovered the cellular viability nor produced antioxidant activity in human astrocytes cell line (CRL-2020 cells) challenged with glutamate neurotoxin for 24 h, but was able to attenuate lipid peroxidation and exerted antiapoptotic and antinecrotic activities [26].Cytoprotective effects of individual tocotrienols isomersSimultaneous treatmentA simultaneous treatment with α-tocotrienol (α-TCT) (0.1, 1 and 10 µM), γ-TCT (1 and 10 µM) and δ-TCT (10 µM) for 24 h enhanced the cellular viability of primary cells of the anterior striatum of foetuses of Wistar rats (17th–19th day of gestation) against hydrogen peroxide-induced neurotoxicity [27]. Similarly, a simultaneous treatment with α-, γ- or δ-TCT (0.1, 1 and 10 µM) for 24 h was able to enhance the cellular viability of primary cells of the anterior striatum of fetuses of Wistar rats (17th–19th day of gestation) against parquet-induced neurotoxicity [27]. Likewise, a 24-h simultaneous treatment with α- and γ-TCT (0.1, 1 and 10 µM), as well as δ-TCT (1 and 10 µM), enhanced the cellular viability of primary cells of anterior striatum of fetuses of Wistar rats (17th–19th day of gestation) against S-nitrosocysteine-induced neurotoxicity [27]. Moreover, a similar observation was made when simultaneous treatment of α-TCT, γ-TCT and δ-TCT, and 3-morpholinosydnonimine was performed [27]. When the same cells were challenged for 48 h with L-buthionine (S,R)-sulfoximine and α-TCT, γ-TCT and δ-TCT also enhanced the cellular viability of primary cells of the anterior striatum of fetuses of Wistar rats through preventing DNA fragmentation [27]. Again, the simultaneous treatment with 10 µM of α-TCT rather than γ- or δ-TCT exerted an apoptotic effect in the cells [27], (Table 6).Furthermore, simultaneous treatment with 10 µM of α-TCT reduced the levels of hydrogen peroxide-induced ROS in human neuroblastoma cells [SH-SY5Y wild-type] [23], although the same treatment was observed to increase the levels of β-amyloid proteins in human neuroblastoma cells overexpressing the human APP695 isoform [SH-SY5Y APP] and those expressing C99 [SH-SY5Y cells] through direct stimulation of the β- and γ-secretase proteolytic enzymes. The same treatment was equally found to induce Aβ degradation in a mouse neuroblastoma cell line (N2a) through inhibiting the insulin-degrading enzyme [23]. However, the direct activation of γ-secretase was independent of the transcription of the presenilin 1 (PSEN1), presenilin 2 (PSEN2), nicastrin (NCSTN), presenilin-enhancer 2 (PSENEN) and anterior-pharynx-defective 1A (APH1A) genes [23]. Interestingly, 10 µM of α-TCT significantly reduced the total cholesterol and free cholesterol in a human neuroblastoma cell line [SH-SY5Y wild-type] [23], (Table 6).Pre-treatmentA five minute pre-treatment with 250 nM α-TCT enhanced cellular viability of mouse hippocampal cells line (HT4) challenged with either glutamate (for 12, 24 or 36 h) or homocysteic acid (for 2, 6, 12 or 24 h) through a direct inhibiting effect on the inducible 12-lipoxygenase enzyme, thereby maintaining neuronal growth [25], preventing the overexpression of c-Src and 2-lipoxigenase enzymes [13] and exerting an antioxidant activity (increasing the ratio of cellular content of reduced glutathione/oxidized glutathione) [13]. Similar to the mouse hippocampal cell line (HT4), the same pre-treatment enhanced the cellular viability of primary cortical neurons of fetuses of Sprague–Dawley rats (17th day of gestation) challenged with homocysteic acid for 24 h [13]. Similar effects were also observed when lower concentrations (25, 50 and 100 nM) of α-TCT were used on the primary cortical neurons of Sprague–Dawley rats challenged with either glutamate or L-homocysteic acid [25]. The effects of other neurotoxin-like L-buthionine (S,R)-sulfoximine alone or L-buthionine (S,R)-sulfoximine plus arachidonic acid were equally attenuated by 100 nM α-TCT [25]. The viability of the cerebral cortex neurons of mouse foetuses (C57BL/6) and B6.129S2-Alox15^tm1Fun^ mice (14th day of gestation) challenged with glutamate, L-buthionine (S,R)-sulfoximine or L-buthionine (S,R)-sulfoximine + arachidonic acid for 24 h were enhanced following pre-treatment with 100 nM of α-TCT [25], (Table 6).Higher doses (0.25 µM) of α-TCT were, however, found not to be effective in attenuating the neurotoxic effects of L-buthionine (S,R)-sulfoximine plus arachidonic acid on mouse hippocampal HT4 neurons, although it was able to attenuate damage from L-buthionine (S,R)-sulfoximine alone [29]. Similarly, a five minute pre-treatment (0.25 µM of α-TCT) of mouse hippocampal neurons (HT4) challenged with L-arachidonic acid for 24 h protected the cells against toxicity, as evidenced by the inhibition of tyrosine phosphorylation of inducible 12-lipoxignase enzyme and the direct inhibition of the inducible 12-lipoxignase enzyme [29]. Additionally, higher concentrations of α-TCT (2.5 and 10 µM) enhanced cellular viability when the cells were challenged with homocysteic acid for 24 h [13]. Moreover, antioxidant activity was potentiated in these cells for up to 6 h after incubation with homocysteic acid mediated via the increase in the ratio of the reduced/oxidized glutathione ratio [13]. Prolonged incubation with the homocysteic acid (8 h) showed a complete elimination of ROS [13]. When the cells were challenged with linoleic acid under similar conditions, they were found to have been protected against the lipid peroxidation and the build-up of ROS [13]. Furthermore, the viability of murine hippocampal HT4 neuronal cells challenged with glutamate for either 30 min or 24 h was found to have increased when the cells were pretreated with 250 µM α-TCT for 10 min or 2 h. These effects were mediated by decreasing the release of arachidonic and docosahexaenoic acids from the cell membrane through inhibiting the hydrolysing effect of cytosolic phospholipase A2 on the cell membrane, which were thought to have then prevented the translocation of cytosolic phospholipase A2 to the cell membrane, its Ser505 phosphorylation or its direct inhibition [24], (Table 6).Post-treatmentWhen hippocampal HT4 neural cells were challenged with homocysteic acid for 24 h and subsequently treated with 250 nM of α-TCT for 8 h, the cellular viability was not improved [13]. Higher concentrations of the α-TCT (0.25, 2.5 and 10 µM), were, however, able to protect the cells from the damaging effects of homocysteic acid [13], (Table 6).

#### 3.5.2. Preclinical Animal Studies

Neuroprotective effects of palm oilOnly one animal study was eligible, and showed that a long-term (8 months) ad libitum feeding on diet enriched with 5 grams of palm oil could slightly enhance the cognitive performance of young (3 week old) male healthy ICR mice, as evidenced by improved spatial learning and memory abilities [33], (Table 7).Neuroprotective effects of tocotrienol-rich fraction (TRF)Healthy animal modelsA single oral daily dose of 100 mg/kg of TRF for 10 weeks exerted a slight increase in the cognitive function of healthy male Wistar rats without inducing an inflammatory effect (normal levels of α-TNF, IL-1β, p56 subunit of NFκβ), apoptosis (normal level of caspase-3) or an alteration in the cholinergic function (normal levels of cholinesterase). Moreover, the redox status was also found to be maintained within healthy limits (normal levels of superoxide dismutase enzyme, catalase enzyme, nitrites and malondialdehyde) in the cerebral cortex and hippocampus of these rats [32]. Conversely, the cognitive function of male progeny of Sprague–Dawley rats was improved significantly when given ad libitum diet-admixed TRF during gestation, lactation and post weaning for 8 weeks [34]. The α-tocotrienol isomer of TRF was found to be highest in the plasma and brain of healthy rats as compared to the other isomers of TRF. Similarly, long-term oral single daily doses of 200 mg/kg TRF in young (3 months) male Wistar rats for 8 months significantly enhanced cognitive function and antioxidant activity (increased activity of superoxide dismutase, catalase and glutathione peroxidase) and decreased DNA damage (higher levels of plasma DNA) [38]. However, the oral single daily dose of 200 mg/kg of TRF for three months did not produce similar results, nor did it alter the serum lipid peroxidation of the young (3 months) healthy male Sprague–Dawley rats [39]. When a similarly high dose of TRF (200 mg/kg) was administered for 3 months in elderly (21 months) healthy rats, the cognitive function, plasma lipid peroxidation, and plasma antioxidant activity were significantly improved, while DNA damage was attenuated [39]. A five-week single oral daily dose (200 mg/kg) of TRF was also shown to improve morphological features in parts of the brain of rats. Accordingly, it induced significant proliferation of granular cells in the dentate gyrus in the hippocampus of chronic stressed or unstressed healthy male Sprague–Dawley rats [37], (Table 7).Disease-induced animal modelsUsing a diabetic model, a single oral daily dose of 50 or 100 mg/kg of α-tocotrienol for 21 days was found to significantly (dose-dependent) enhance cognitive function, attenuate brain lipid peroxidation (decreased level of malondialdehyde (MDA)) and enhance brain antioxidant activity (increased levels of reduced glutathione (GSH), superoxide dismutase (SOD) and catalase (CAT)). Similarly, the brain cholinergic function was slightly enhanced (non-significant dose-dependent reduction in the level of brain acetyl cholinesterase enzyme) [36]. When doses of 25, 50 or 100 mg/kg of TRF were given for 10 weeks, the cognitive function was enhanced in these rats, as well as the cerebrocortical and hippocampal cholinergic function, lipid peroxidation (reduced level of MDA), antioxidant activity (increased SOD and CAT), inflammation (reduced levels of TNF-α, IL-1β and p56 subunit of NFκβ) and antiapoptotic effects (reduced level of cerebrocortical and hippocampal levels of caspase-3) [32], (Table 7).Transgenic animal modelsUsing a double transgenic Alzheimer’s disease (AβPP/PS1) animal model, a long-term single oral daily dose of 60 mg/kg of TRF for 10 months in male mice (5 months of age) significantly enhanced recognition abilities and reduced the deposition of β-amyloid proteins in the cortex and hippocampus (soluble and insoluble Aβ isoforms: Aβ _40_, Aβ _42_ and Aβ oligomer) [30]. Similarly, the same single oral daily dose of TRF (60 mg/kg) for the same duration (10 months) in AβPP/PS1 male mice (5 months of age) slightly enhanced cognitive function, and metabolomics analyses indicated an alteration in 90 putative metabolites that are involved in several metabolic Alzheimer’s disease pathways [31]. In addition, a single oral daily dose of 200 mg/kg of TRF for 6 months in AβPP/PS1 male mice (aged 9 months) upregulated the genes that are responsible for neuroprotective effects, such as Slc24a2 (solute carrier family 24 [sodium/potassium/calcium exchanger]), exo1 (exonuclease 1), and Enox1 (ecto-NOX disulfide-thiol exchanger 1), and downregulated the genes responsible for the pathology of AD, such as Pla2g4a (phospholipase A2, group IVA [cytosolic, calcium-dependent]), Tfap2b (transcription factor AP-2 beta) [35], (Table 7).

## 4. Discussion

The consumption of palm oil is increasing globally; however, presently, it is a highly controversial food. Palm oil is used for cooking and is also added to many ready-to-eat foods in grocery stores. There is a lot of interest and controversy surrounding its consumption [40]. It is the most produced oil globally, with the cheapest price on the market, and is the most stable against oxidation [41]. It has been linked to several health benefits, including protecting brain functions, reducing heart disease risk and improving vitamin A status [42]. A deficiency of vitamin E is associated with the occurrence of several neurological disorders, while preventing vitamin E deficiency in vulnerable persons requires large quantities of vitamin E, often more than the RDA [43]. This has prompted a strong justification for the consumption of palm oil, since it is one of the richest sources of vitamin E isomers (tocopherols and tocotrienols), and the loss of vitamin E isomers from palm oil after deep frying was reported to be minimal (8%) [43]. Moreover, the end product of palm oil after refining still retains a considerable concentration of vitamin E [44], mostly in the form of tocotrienols (70%), but also some tocopherols (30%) [45]. TRF contains both tocopherols and tocotrienols isomers (α-, β-, γ- and δ) [31], as 70% tocotrienols and 30% tocopherols [30]. Tocotrienols have been reported to exert more powerful antioxidant activities than tocopherols [36]. The reducing abilities of tocotrienol isomers decreases in the order of α > β > γ > δ [46]. TRF has been reported to be beneficial in several pathologies, including cancer, diabetes mellitus, aging, and neurodegeneration [35], while the α-tocotrienol isomer constitutes the highest abundancy in TRF, making it more effectively cytoprotective than the other tocotrienol isomers of TRF [27]. The former reported findings could indicate that the α-tocotrienol isomer is the one which is responsible for the biological activities of TRF, particularly due to the fact that nanomolar contractions of α-tocotrienols exerted a potent neuroprotective effect [24].

The neuroprotective properties of palm oil and its bioactives are of great interest to consumers, researchers, health authorities and policymakers worldwide, since palm oil is widely consumed [11]. Hence, this systematic review represents an attempt at conducting a risk–benefit analysis of the consumption of palm oil, in particular, to critically analyse the neuroprotective effects of palm oil and its bioactives in neuronal cells and animal models and further explore the possibilities of developing therapeutic agents from palm oil bioactives for the treatment of prophylaxis against neurotoxicity.

In animal models, a marginal increase in the cognitive performance of mice was observed [33], which may be explained by the low quantity of palm oil relative to the diets used for the intervention. The quantities used translate to low concentrations of tocotrienols that may not exert a significant effect on cognitive performance since tocotrienols have been shown to be responsible for the neuroprotective effect of palm oil [47]. In addition, future evaluation of the effect of palm oil on cognitive function should consider the functional and structural changes, as well as the underlying molecular mechanisms underlying cognitive function, using long-term multilevel interventions of palm oil as well as animals with different ages (young, adults and elderly) and sexes (males and females).

Acetylcholinesterase is an enzyme that catalyzes the breakdown of acetylcholine [48], the decline of which contributes to the development of dementia and neurodegeneration. [49], which suggested that the cognitive supportive effect of TRF in healthy rats was independent of the central cholinergic function since the levels of acetylcholinesterase enzyme of these rats were still normal. The resulting marginal enhancement in cognitive function of healthy rats given 100 mg/kg of TRF [32] could be due to either the poor penetration of the tocotrienol isomers in TRF through the blood–brain barriers, since blood–brain barriers act to restrict the penetration of several molecules into the brain [50], or the dose of TRF (100 mg/kg) and the duration of treatment (6 weeks) were insufficient to induce a pronounced enhancement of cognitive function in healthy rats [32], particularly when it is noted that 200 mg/kg of TRF for a longer duration (8 months) significantly enhanced the cognitive functions and antioxidant activity in healthy rats [38]. However, the ad libitum 100 mg/kg of diet-admixed TRF significantly enhanced the cognitive function of young rats [34]. However, the oral daily dose of 200 mg/kg of TRF in healthy stressed and unstressed rats for 5 weeks did not alter the proliferation and survival of granular cells in the dentate gyrus of the hippocampus [37], which could support the hypothesis of the poor penetration of tocotrienols through the blood–brain barriers, depending on the maturity of blood–brain barriers between adults and natal animals and the situation of the health status of the blood–brain barriers [50].

Diabetes mellitus is a metabolic disorder [51] and is the seventh leading cause of death worldwide [52]. Uncontrolled sustained hyperglycemia is the main pathophysiological feature of type 1 and type 2 diabetes mellitus, leading to multisystemic complications [53]. Recent evidence suggests mutual underlying pathologies between diabetes mellitus and neurodegeneration [54] due to the homeostatic imbalance of glucagon–insulin in diabetes mellitus, which induces amyloid deposition in the pancreatic islets of Langerhans’s and the brain [53]. In addition, diabetes mellitus and neurodegeneration share common pathophysiological events, including inflammation and oxidative stress [54]. The improved cholinergic function in diabetic rats given TRF was achieved through reducing the cerebrocortical level of acetyl cholinesterase [32]—the enzyme that catalyzes the breakdown of acetylcholine neurotransmitter [48]—the decline of which contributes to the development of dementia and neurodegeneration [49]. In addition, TRF attenuated cerebrocortical and hippocampal oxidative stress and inflammation [32], which are mutual features of the pathophysiology of neurodegeneration and diabetes mellitus [54]. Moreover, TRF exerted an antiapoptotic effect, thereby enhancing the survival of cerebrocortical and hippocampal neurons [32]. The attenuation of oxidative stress by α-tocotrienol also suggests that it could benefit both diabetes mellitus and neurodegeneration [54]. However, the study did not demonstrate the success of inducing diabetes by measuring baseline and final blood sugar level and/or insulin [36] since α-tocotrienol possesses an antioxidant activity [47] that could provide protection against the destructive effect of streptozotocin (STZ) on pancreatic β-cells. Moreover, the sample size of animals per group was not specified. Furthermore, while the study was designed to be a chronic study, the duration of intervention was only 21 days [36]. Eventually, TRF exerted a potent antioxidant activity and was composed of four tocotrienol isomers (α, β, γ and δ) [45], of which the reducing ability decreased in the order of α > β > γ > δ [46]. Therefore, the similarly enhanced cognitive functions and antioxidant activity of diabetes-induced rats by α-tocotrienol and TRF in the studies by Tiwari et al. [36] and Kuhad et al. [32], respectively, could indicate that α-tocotrienol is the isomer that is responsible for the improved cognitive function and antioxidant activities of TRF.

The brains of patients with Alzheimer’s disease demonstrate the presence of fibrillar amyloid-β peptide [55]. In addition, three gene mutations are responsible for Alzheimer’s disease, including presenilin 1 (PSEN1), presenilin 2 (PSEN2) and amyloid precursor protein (APP), with presenilin 1 being the most common [56]. Hence, heterozygous AβPP/PS1 double transgenic male mice were used since they represent human APP and human PS1. Although the functional outcomes indicated that TRF (60 mg/kg) for 10 months could not enhance the cognitive functions or reduce the deposition of soluble and insoluble Aβ isomers in the cortex and hippocampus of AβPP/PS1 male mice, the structural outcomes indicated a relative reduction in the deposition of Aβ-protein and Aβ-plaques significantly in the cortex of animals but non-significantly in hippocampus [30]. Also, the same dose of TRF (60 mg/kg) for the same duration (10 months) in the same animals (AβPP/PS1 male mice) indicated a slight enhancement of the cognitive function; however, metabolomics indicated that TRF induced 90 putative metabolites that are involved in several metabolic pathways in Alzheimer’s disease [31]. However, a higher dose of TRF (200 mg/kg) conferred neuroprotective effects through a significant up-regulation of Slc24a2 (solute carrier family 24 [sodium/potassium/calcium exchanger]), exo1 (exonuclease 1) and Enox1 (ecto-NOX disulfide-thiol exchanger 1). Additionally, Pla2g4a (phospholipase A2, group IVA [cytosolic, calcium-dependent]), and Tfap2b (transcription factor AP-2 beta) were significantly down-regulated [35]. It is, therefore, our belief that several dose levels (in between 60 and 200 mg/kg as well as higher than 200 mg/kg of TRF) would be better when assessing dose–response effects [57], particularly relating to the neuroprotective effect that TRF provides, as evidenced by the findings of the current systematic review. In addition, it would be better to elucidate the efficacy of the individual tocotrienol isomers of TRF in AβPP/PS1 double transgenic male mice to evaluate which tocotrienol isomer is responsible for the neuroprotective effect of TRF and the structure–activity relationships, so that these isomers could be recruited as prototype molecules to develop analogues of higher efficacy and effectiveness.

In in vitro studies, the neuroprotective effects of simultaneous treatment with TRF against the neurotoxicity of hydrogen peroxide [27] and Aβ42 aggregates [30] was conflicting, although the duration of exposure was the same (24 h). Nonetheless, hydrogen peroxide exerted neurotoxicity through the generation of oxygen free radicals [27], which were neutralized by TRF [58]. Conversely, the inability of TRF to improve cellular viability in the presence of β-amyloid protein (Aβ42 aggregates) in the human neuroblastoma cell line (SH-SY5Y) as reported by Grimm et al. [23] suggests that α-tocotrienol is the bioactive isomer responsible for the activity of TRF. Future in vivo studies should verify the neuroprotective effects of different concentrations of either TRF or tocotrienol isomers against the neurotoxic effects of amyloid-β protein, especially since the efficacy of TRF in AβPP/PS1 mice is inconclusive [30]. Furthermore, the reduced total and free cholesterol in the human neuroblastoma cells line [SH-SY5Y wild-type] due to α-tocotrienol treatment [23] suggests that α-tocotrienol could provide a potent neuroprotective effect in Alzheimer’s disease, since the elevated serum levels of cholesterol are positively correlated with dementia and Alzheimer’s disease [59].

Glutamate is a highly abundant neurotransmitter in the central nervous system (CNS), which, when accumulated, could result in neurodegeneration [60] and neuronal death [61]. TRF had prophylactic and recovery neuroprotective effects on neurons treated with glutamate, mediated via the attenuation of lipid peroxidation and apoptosis [28]. TRF also directly inhibits glutamate-inducible 12-lipoxignase enzyme [29], an enzyme that is involved in the pathogenicity of Alzheimer’s disease through lipid peroxidation of the cell membrane of neurones [62]. Both pre-treatment and post-treatment with TRF protected the integrity of cell membranes [28]; however, the underlying molecular mechanism was not evaluated. Conversely, the effect of TRF was not as effective in astrocytes as in neurons [26]. Accordingly, further in vivo studies should be conducted to elucidate the antioxidant, antiapoptotic and antinecrotic effects of TRF on neuronal and neuroglial cells of animals, or even in vitro studies using human neuronal and neuroglial cell lines.

TRF is a mixture of several tocotrienols and tocopherols [47]; therefore, it is very important to elucidate the neuroprotective effects of individual tocotrienols, since the effects of tocopherols have been widely evaluated [23,27]. The findings of simultaneous treatment indicated that α-, γ- and δ-tocotrienols exerted a substantial neuroprotective effect against a wide range of neurotoxins, which could be due to their antioxidant and antiapoptotic activities [27]. However, the simultaneous neuroprotective effect of α-tocotrienol was superior to those of either γ- and δ-tocotrienols [27], which could be related to the reducing abilities of tocotrienol isomers that decrease in the order of α > β > γ > δ [46]. Although simultaneous treatment with α-tocotrienol enhanced cellular survival, it exerted an amyloidogenic effect by increasing the levels of β-amyloid through a direct activation of β- and α-secretase enzymes [23]. The majority of studies indicated that pre-treatment of neuronal cells with nanomolar concentrations (25, 50, 100 and 250 nM) or micromolar concentrations of α-tocotrienol [13,24,25,29] could provide substantial prophylactic neuroprotective activity for neuronal cells against a wide range of neurotoxins. The effects were found to be medicated through exerting antioxidant activity and maintaining the integrity of the cell membrane of neurones by inhibiting the hydrolytic activity of phospholipase A2 and lipoxygenase enzymes on the phospholipids of the cell membranes [13,24,25,29]. Future preclinical in vitro and in vivo studies should consider the elucidation of the pre-treatment effect of tocotrienols on the levels of AB protein isomers, since simultaneous treatment with α-tocotrienols was reported to be amyloidogenic [23]. On the other hand, post-treatment with nanomolar and micromolar concentrations of α-tocotrienol showed neuroprotective effects [13].

Finally, this systematic review recommends that further studies should undertake the practice of bias as well as baseline characteristics and identical exposure characteristics during the allocation of animals or cells, measurement of outcomes, adequate addressing of attrition, analysing outcomes and reporting results [21,22]. In the literature, little was reported about the neuroprotective effects of the organic acids (e.g., palmitic acid) in palm oil. Accordingly, this systematic review recommends that further investigations should consider the investigation of the neuroprotective effects and the underlying molecular mechanisms of the oleic fractions and organic acids (e.g., palmitic acid, oleic acid) in palm oil, because the oleic fraction constitutes the majority of palm oil.

## 5. Conclusions

In animal studies, palm oil, TRF and α-tocotrienol isomer enhanced the cognitive performance of healthy animals, while TRF and α-tocotrienol isomer enhanced cognitive functions with an attenuation of neuro-oxidative stress, neuroinflammation and neuro-apoptosis in a diabetes-induced and transgenic AD animal model. In transgenic AD animal models, TRF also marginally reduced the deposition of amyloid-β isomers. Furthermore, results from in vitro cell studies showed that TRF and α-tocotrienol exerted prophylactic neuroprotective effects, while α-tocotrienol exerted recovery neuroprotective effects. In addition, α-, γ- and δ-tocotrienol isomers exerted neuroprotective effects in neuronal cells; however, α-tocotrienol was superior to γ- and δ-tocotrienol isomers. Although α-tocotrienol isomer provided a potent reduction in cholesterol levels, it was associated with amyloidogenic activity. However, the findings of the disposition of amyloid-β proteins in the preclinical cell and animal studies have been conflicting. Our systematic review infers that the activity of TRF from palm oil could be attributed mostly to its α-tocotrienol content.

## Figures and Tables

**Figure 1 nutrients-12-00521-f001:**
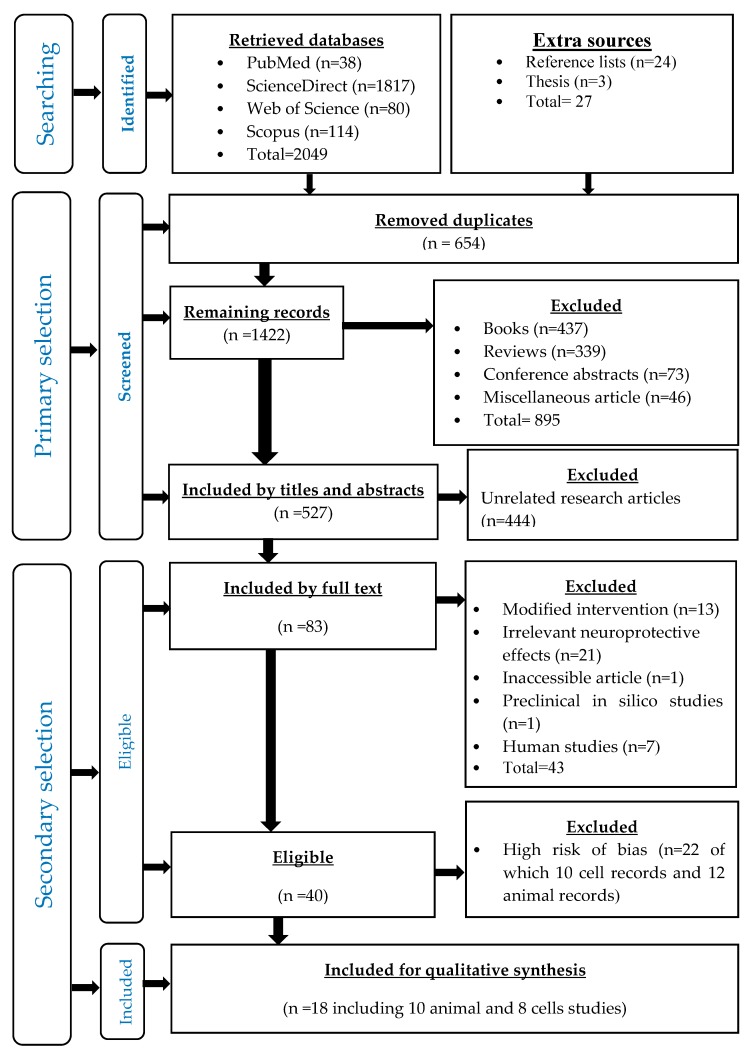
Flow chart of all stages of the systematic review according to PRISMA statement.

**Table 1 nutrients-12-00521-t001:** Inclusion and exclusion criteria of eligibility of preclinical cell studies.

		Inclusion Criteria	Exclusion Criteria
**Human disease model**		●Neurotoxicity	●irrelevant
		●neurodegeneration	
		●neuro-apoptosis	
		●neuro-oxidative stress	
		●neuro-inflammation	
**Population**		●normal or transgenic neuronal cell line	●neuronal or neuroglial cell line derived from an organism with a neurological hereditary disease ●primary neuronal or neuroglial cells derived from an organism with a neurological hereditary disease; ●in silico models
		●normal or transgenic neuroglial cell line	
		●primary neuronal or neuroglial cells	
		●neuronal slices	
**Interventions**		●palm oil and palm oil bioactives (tocotrienol-rich fraction, polyphenol-rich fraction, individual tocotrienols or β-carotenes)	●pure α-, β-, δ- or γ-tocopherols
		●any duration of intervention	
		●any dose of intervention	
		●any timing of intervention (simultaneous treatment: incubation of the intervention and the neurotoxin at the same time)	
		●pre-treatment: cells treated before being challenged with the neurotoxin	
		●post-treatment: cells treated after being challenged with the neurotoxin	
**Comparators**		●inert vehicles (water, ethanol, normal saline, phosphate buffer, DMSO or media)	●comparator with different experimental conditions or exposure compared with the intervention groups
		●comparators subjected to identical experimental conditions and exposure as the intervention groups	●tocopherols
		●the same vehicle used to dissolve the intervention	●a vehicle rather than that used to dissolve the intervention
**Study Design**		●Preclinical in vitro experiments with mono-level or multi-level intervention (pre-, post- and simultaneous exposure) with an appropriate comparator	●lack of an appropriate comparator
**Outcomes**	**Primary**	●Cellular viability	●irrelevant
		●inflammation	
		●apoptosis;	
		●oxidative stress (lipid peroxidation and antioxidant activity)	
	**Secondary**	●Cytomorphological and molecular changes	●irrelevant
**Others (article type)**		●published research articles ●full papers in proceedings ●unpublished theses	●research articles in predatory journals according to Beall’s list
			●Published theses
			●inaccessible research articles
			●high-risk biased studies

**Table 2 nutrients-12-00521-t002:** Inclusion and exclusion criteria of eligibility of preclinical animal studies.

		Inclusion Criteria	Exclusion Criteria
**Human disease model**		●neurodegenerative disorders	●irrelevant
		●neuroinflammation	
		●neurotoxicity	
		●neuro-injury	
		●neuro-oxidative stress	
**Population**		●pup, young, young adult, adult or elderly animals	●none
		●male or female animals	
		●rats or mice	
		●strains of rats or mice	
		●healthy, disease-induced animals or transgenic disease animals	
**Interventions**		●palm oil, or its bioactives (e.g., palmitic acid, tocotrienol-rich fraction, polyphenol-rich fraction, individual tocotrienols or β-carotenes)	●pure α-, β-, δ- and γ-tocopherols ●blended palm oil with other oils ●palm oil combined with other foods ●content-modified palm oil (e.g., vitamin-E-stripped) ●palm oil bioactives extracted from parts of palm tree other than palm fruit
		●any dose of intervention	
		●any timing of intervention	
		●any frequency of intervention (e.g., once or twice… etc. per a day)	
		●any duration of intervention	
		●any technique of intervention administration (admixed with diet, suspended in water, oral via gastric gavage or parenteral)	
**Comparators**		●palm oil; inert vehicles (water, normal saline, or tweens) ●comparators subjected to identical experimental conditions and exposure similar to those of the intervention groups ●the same vehicle used to dissolve the intervention	●blended palm oil with other oils
			●palm oil combined with other foods; content-modified palm oil (e.g., vitamin-E-stripped)
			●comparator with different experimental conditions or exposure different from the intervention groups
			●a vehicle rather than that used to dissolve the intervention
**Study Design**		●acute, sub-acute or chronic preclinical animal studies containing at least mono-level or multi-level dosing of oral dietary, oral gavage or parenteral intervention with an appropriate comparator	●lack an appropriate comparator
**Outcomes**	**Primary**	●cognitive function	●irrelevant
		●locomotor function	
		●healing after neuro-injury	
		●neuroinflammation	
		●apoptosis	
		●oxidative stress (lipid peroxidation and antioxidant activity)	
	**Secondary**	●structural and molecular changes	●irrelevant
**Others (article type)**		●published research articles ●full papers in proceedings ●Unpublished theses	●research articles in predator journals according to Beall’s list
			●Published theses
			●inaccessible research articles
			● high-risk biased studies

**Table 3 nutrients-12-00521-t003:** Number of results per hit in each database.

Keywords	PubMed	Web of Science	Science Direct	Scopus
Palm oil and nervous system	17	9	499	12
Palm oil and brain	15	64	1071	83
Palm oil and neurodegenerative diseases	4	5	135	7
Palm oil and cognition	2	2	112	12
**Total**	**38**	**80**	**1817**	**114**

**Table 4 nutrients-12-00521-t004:** Assessment of risk of bias for the preclinical cell studies using the OHAT tool.

Studies	Selection Bias		Performance Bias		Detection Bias			Attrition Bias	Reporting Bias	Others
	Randomization and Concurrent Control Group	Allocation Concealment	Identical Vehicle	Blinding	Accuracy of Exposure Characterization	Consistent Exposure Administration	Blinding	Incomplete Outcome Data	Selective Outcome Reporting	Other sources of Bias
[23]	PL	NR	DL	NR	PL	DL	NR	NR	PL	PL
[24]	PL	NR	DL	NR	PL	DL	NR	NR	PL	PL
[13]	PL	NR	DL	NR	PL	DL	NR	NR	PL	PL
[25]	PL	NR	DL	NR	PL	DL	NR	NR	PL	PL
[26]	PL	NR	PL	NR	DL	DL	NR	NR	PL	PL
[27]	PL	PL	DL	DL	DL	DL	PL	NR	DL	PL
[28]	PL	NR	DL	NR	PL	DL	NR	NR	PL	PL
[29] *	PL	NR	DL	NR	PL	DL	NR	NR	PL	PL
[30]	PL	NR	PL	NR	PL	DL	NR	NR	PL	PL

DL (Definitely Low risk of bias if direct evidence of low risk-of-bias practices), PL (Probably Low risk of bias: Indirect evidence of low risk-of-bias practices OR it is deemed that deviations from low risk-of-bias practices for these criteria during the study would not appreciably bias results, including consideration of direction and magnitude of bias), PH (Probably High risk of bias: Indirect evidence of high risk-of-bias practices OR there is insufficient information “NR”) and DH (Definitely High risk of bias: Direct evidence of high risk-of-bias practices). * superscript: This study was a part of a preclinical animal study.

**Table 5 nutrients-12-00521-t005:** Assessment of risk of bias for the preclinical animal studies using CYRCL’s tool.

Studies	Selection Bias			Performance Bias		Detection Bias		Attrition Bias	Reporting Bias	Others
	Random Sequence Generation	Baseline Characteristics	Allocation Concealment	Random Housing	Blinding	Random Outcome Assessment	Blinding	Incomplete Outcome Data	Selective Outcome Reporting	Other sources of Bias
[31]	No	Yes	U	No	No	Yes	No	Yes	Yes	Yes
[30]	No	Yes	U	No	No	No	No	Yes	Yes	U
[32]	Yes	Yes	U	No	No	Yes	No	No	Yes	U
[33]	Yes	Yes	U	No	No	No	No	Yes	Yes	U
[34]	Yes	Yes	U	No	No	No	No	Yes	Yes	U
[35]	Yes	Yes	U	No	No	Yes	No	No	Yes	Yes
[36]	Yes	Yes	U	Yes	No	No	No	No	yes	U
[37]	Yes	Yes	U	No	No	No	Yes	No	Yes	U
[38]	Yes	Yes	U	No	No	Yes	No	Yes	Yes	U
[39]	Yes	Yes	U	No	No	Yes	No	No	Yes	U

“Yes” to indicate a low risk of bias, “No” to indicate a high risk of bias or “U” to indicate an uncertain level of bias.

**Table 6 nutrients-12-00521-t006:** Characteristics of preclinical cells studies.

Reference	Study Design, Human Disease Modelled and Population	Intervention	Comparator	Outcomes	
				Primary	Secondary
[25]	●Glutamate induced-neurotoxicity model for 12, 24 or 36 h ●Mouse hippocampal HT4 cells line	●5-min pre-treatment with 250 nM of α-TCT in 1% ethanol	●1% ethanol	●A significant time-dependent enhancement of cellular viability	●Direct inhibition of inducible 12-lipoxygenase enzyme. ●Morphological changes indicated the prevention of neurodegeneration with the maintenance of neuronal growth.
[25]	●Glutamate or L-homocysteic acid neurotoxicity for 24 h ●Immature primary cortical neurons of Sprague-Dawley rats (17th day of gestation)	●5-min pre-treatment with 25, 50, 100 and 250 nM of α-TCT in 1% ethanol	●1% ethanol	●A significant enhancement of cellular viability	
[25]	●L-buthionine (S,R)-sulfoximine or L-buthionine (S,R)-sulfoximine +arachidonic acid neurotoxicity for 24 h using immature primary cortical neurons of Sprague-Dawley rats (17th day of gestation)	●5-min pre-treatment with 100 nM of α-TCT in 1% ethanol	●1% ethanol	●A significant enhancement cellular viability	
[25]	●L-buthionine (S,R)-sulfoximine neurotoxicity for 24 h using immature primary cortical neurons of Sprague-Dawley rats (17th day of gestation)	●5-min pre-treatment with 100 nM of α-TCT in 1% ethanol	●1% ethanol	●A significant enhancement of cellular viability, but loss of the cellular reduced glutathione	
[25]	●Glutamate, L-buthionine (S,R)-sulfoximine or L-buthionine (S,R)-sulfoximine + arachidonic acid neurotoxicity for 24 h using cerebral cortex neurons of mouse fetuses (C57BL/6) mice, (14th day of gestation)	●5-min pre-treatment with 100 nM of α-TCT in 1% ethanol	●1% ethanol	●A significant enhancement of cellular viability	
[25]	●Glutamate, L-buthionine (S,R)-sulfoximine or L-buthionine (S,R)-sulfoximine + arachidonic acid for 24 h using cerebral cortex neurons of the fetuses of B6.129S2-Alox15tm1Fun mice	●5-min pre-treatment with 100 nM of α-TCT in 1% ethanol	●1% ethanol	●A significant enhancement of cellular viability	
[27]	●Hydrogen peroxide neurotoxicity for 24 h using primary cells of anterior striatum of fetal Wistar rats (17th–19th day of gestation).	●Simultaneous treatment with 0.1, 1 or 10 µM of TRF in 0.1% DMSO (TRF: 90% pure contains 14.5. mg α-TCT, 2.5 mg β-TCT, 26 mg γ-TCT and 7.2 δ-TCT)	●0.1% DMSO	●A significant enhancement of cellular viability.	
[27]	●Hydrogen peroxide neurotoxicity for 24 h using primary cells of anterior striatum of fetal Wistar rats (17th–19th day of gestation)	●Simultaneous treatment with 0.1, 1 or 10 µM of either α-, γ- or δ-TCT in 0.1% DMSO	●0.1% DMSO	●α-TCT [0.1, 1 and 10 µM], γ-TCT [1 and 10 µM] and δ-TCT [10 µM] significantly enhanced cellular viability	
[27]	●Parquet neurotoxicity with for 24 h using primary cells of anterior striatum of foetal Wistar rats on the 17th–19th day of gestation	●Simultaneous treatment with 0.1, 1 and 10 µM of either α-, γ- or δ-TCT in 0.1% DMSO	●0.1% DMSO	●α-, γ- or δ-TCT [0.1, 1 and 10 µM] significantly enhanced cellular viability	
[27]	●S-nitrosocysteine neurotoxicity for 24 h using primary cells of anterior striatum of foetal Wistar rats on the 17th–19th day of gestation	●Simultaneous treatment with 0.1, 1 and 10 µM of either α-, γ- or δ-TCT in 0.1% DMSO	●0.1% DMSO	●α- and γ-TCT [0.1, 1 and 10 µM] as well as δ-TCT [1 and 10 µM] significantly enhanced cellular viability.	
[27]	●3-morpholinosydnonimine neurotoxicity for 24 h using primary cells of anterior striatum of foetal Wistar rats on the 17th–19th day of gestation	●Simultaneous treatment with 0.1, 1 and 10 µM of either α-, γ- or δ-TCT in 0.1% DMSO	●0.1% DMSO	●α-TCT [0.1, 1 and 10 µM], γ-TCT [1 and 10 µM] and δ-TCT [1 and 10 µM] significantly enhanced cellular viability.	
[27]	●L-buthionine (S,R)-sulfoximine neurotoxicity for 48 h using primary cells of anterior striatum of foetal Wistar rats on the 17th–19th day of gestation	●Simultaneous treatment with 0.01, 0.1 and 1 µM of either α-, γ- or δ-TCT in 0.1% DMSO	●0.1% DMSO	●α-TCT [0.1 and 1 µM], γ-TCT [1 µM] and δ-TCT [1 µM] significantly enhanced cellular viability. α-, γ- and δ-TCT [1 µM] exerted antiapoptotic effects, however, the antiapoptotic effect of α-TCT was superior to that of either γ- or δ-TCT	●Antiapoptotic effect involved the prevention of DNA fragmentation.
[27]	●Staurosporine neurotoxicity for 24 h using primary cells of anterior striatum of foetal Wistar rats on the 17th–19th day of gestation.	●Simultaneous treatment with 10 µM of either α-, γ- or δ-TCT 0.1% DMSO	●0.1% DMSO	●Only 10 µM of α-TCT exerted a significant antiapoptotic effect, while γ- or δ-TCT field to exert a significant antiapoptotic effect.	●Antiapoptotic effect involved a significant prevention of DNA fragmentation.
[29]	●Glutamate neurotoxicity for 24 h using mouse Hippocampal HT4 Neurons	●5-min pre-treatment with 250 nm of TRF in 1% ethanol (TRF: 90% pure contains 14.5. mg α-TCT, 2.5 mg β-TCT, 26 mg γ-TCT and 7.2 δ-TCT)	●1% ethanol	●A significant enhancement of cellular viability	●Inhibiting the tyrosine phosphorylation of inducible 12-lipoxignase enzyme and direct inhibition of inducible 12-lipoxignase enzyme
[29]	●Glutamate neurotoxicity for 24 h using cerebral cortex neurons of foetuses of Sprague-Dawley rats, (17th day of gestation)	●5-min pre-treatment with 250 nm of TRF in 1% ethanol (TRF: 90% pure contains 14.5. mg α-TCT, 2.5 mg β-TCT, 26 mg γ-TCT and 7.2 δ-TCT)	●1% ethanol	●A significant enhancement of cellular viability	●Inhibiting the tyrosine phosphorylation of inducible 12-lipoxignase enzyme and direct inhibition of inducible 12-lipoxignase enzyme
[29]	●L-buthionine (S,R)-sulfoximine neurotoxicity for 24 h using mouse Hippocampal HT4 Neurons	●5-min pre-treatment with 0.25 µM of α-TCT in 1% ethanol	●1% ethanol	●A relative (nonsignificant) enhancement of cellular viability	
[29]	●L-buthionine (S,R)-sulfoximine + arachidonic acid neurotoxicity for 24 h using mouse Hippocampal HT4 Neurons	●5-min pre-treatment with 0.25 µM of α-TCT in 1% ethanol	●1% ethanol	●A significant loss of cellular viability	
[29]	●L- arachidonic acid neurotoxicity for 24 h using mouse Hippocampal HT4 Neurons	●5-min pre-treatment with 0.25 µM of α-TCT in 1% ethanol	●1% ethanol		●Inhibiting tyrosine phosphorylation of inducible 12-lipoxignase enzyme and direct inhibition of inducible 12-lipoxignase enzyme
[13]	●Homocysteic acid neurotoxicity for 24 h using mouse hippocampal HT4 neural cells	●5 min pre- or 8 h post-treatment with 250 nM of α-TCT in 1% ethanol	●1% ethanol	●Pre-treatment significantly enhanced cellular viability, while post-treatment failed to enhance cellular viability	
[13]	●Homocysteic acid neurotoxicity for 24 h using mouse hippocampal HT4 neural cells	●5-min pre- or 8 h post-treatment with 0.25, 2.5 and 10 µM of α-TCT in 1% ethanol	●1% ethanol	●Pre- and post-treatment significantly enhanced cellular viability.	
[13]	●Homocysteic acid neurotoxicity for 2 or 6 h using mouse hippocampal HT4 neural cells	●5-min pre-treatment with 250 nM of α-TCT in 1% ethanol	●1% ethanol	●Provided a significant antioxidant activity through enhancing the ratio of cellular levels of reduced glutathione/oxidized glutathione	
[13]	●Homocysteic acid neurotoxicity for 8 h using mouse hippocampal HT4 neural cells	●5-min pre-treatment with 2.5 and 10 µM of α-TCT in 1% ethanol	●1% ethanol		●Blue fluorescence imaging indicated a completely elimination of ROS
[13]	●Linoleic acid neurotoxicity for 4 h using mouse hippocampal HT4 neural cells	●5-min pre-treatment with 0.25, 1, 2.5 and 10 µM of α-TCT in 1% ethanol	●1% ethanol	●1, 2.5 and 10 µM of α-TCT significantly attenuated lipid peroxidation	●Fluorescence imaging indicated the attenuation of the build-up of ROS
[13]	●Linoleic acid neurotoxicity for 24 h using mouse hippocampal HT4 neural cells	●5-min pre-treatment with 0.25, 1, 2.5 and 10 µM of α-TCT in 1% ethanol	●1% ethanol	●Significantly enhanced cellular viability [2.5 and 10 µM]	
[13]	●Homocysteic acid neurotoxicity for 12 h using mouse hippocampal HT4 neural cells	●5-min pre-treatment with 250 nM of α-TCT in 1% ethanol	●1% ethanol	●A significant enhancement of cellular viability	●Prevented overexpression of c-Src and 2-lipoxigenase
[13]	●Homocysteic acid neurotoxicity for 6 h using mouse hippocampal HT4 neural cells	●5-min pre-treatment with 0.25, 1, 2.5 and 10 µM of α-TCT 1% ethanol	●1% ethanol	●Provided a significant antioxidant activity [2.5 and 10 µM] through enhancing the ratio of cellular levels of reduced glutathione/oxidized glutathione	
[13]	●Homocysteic acid neurotoxicity for 24 h using primary cortical neurons of foetuses of Sprague–Dawley (17th day of gestation)	●5-min pre-treatment with 250 nM of α-TCT in 1% ethanol	●1% ethanol	●Significantly enhanced cellular viability	
[13]	●Homocysteic acid neurotoxicity for 24 h using primary cortical neurons of foetuses of Sprague–Dawley (17th day of gestation)	●5-min pre-treatment with 0.25, 1, 2.5 and 10 µM of α-TCT in 1% ethanol	●1% ethanol	●Significantly enhanced cellular viability	
[24]	●Glutamate neurotoxicity for 30 min using murine hippocampal HT4 neuronal cells	●10-min pre-treatment with 250 µM α-TCT in ethanol 1%	●1% ethanol	●A significant enhancement of cellular viability	●Decreasing significantly the release of arachidonic and docosahexaenoic acids from cell membrane through attenuating the hydrolysis activity of cytosolic phospholipase A2 on cell membrane due to inhibiting: ●Translocation of cytosolic phospholipase A2 to cell membrane, ●Ser505 phosphorylation of cytosolic phospholipase A2 ●Phospholipase A2 activity
[24]	●Glutamate neurotoxicity for 24 h using murine hippocampal HT4 neuronal cells	●2-h pre-treatment with 250 µM α-TCT in ethanol 1%	●1% ethanol	●A significant enhancement of cellular viability	●Direct inhibition of phospholipase A2.
[28]	●Glutamate neurotoxicity for 24 h using human neuroblastoma cells line (SK-N-SH)	●5-min pre-treatment with 100, 200, or 300 ng/mL of TRF in DMSO (TRF: 25% tocopherol and 75% tocotrienols)	●DMSO	●A significant enhancement of cellular viability particularly 200 ng/mL ●A significant dose-dependent attenuation of lipid peroxidation through reducing the levels of MDA	●Annexin V-FITC/PI staining indicated that 200 mg/kg was significantly the highest against necrosis as well as early and late stage apoptosis
[28]	●Glutamate neurotoxicity for 24 h using human neuroblastoma cells line (SK-N-SH)	●30-min post-treatment with 100, 200, or 300 ng/mL TRF in DMSO (TRF: 25% tocopherol and 75% tocotrienols)	●DMSO	●A significant enhancement of cellular viability particularly 200 mg/kg. ●A significant attenuation of lipid peroxidation through reducing the levels of MDA particularly 300 mg/kg	●Annexin V-FITC/PI staining indicated slight (nonsignificant) antiapoptotic effect against necrosis as well as early and late stage apoptosis ●Electronic microscope scanning for cellular morphology indicated that only 200 mg/kg could provide a little improvement to the cell membrane integrity.
[23]	●Hydrogen peroxide neurotoxicity for 24 h using human neuroblastoma cells line [SH-SY5Y wild-type]	●Simultaneous treatment with 10 µM of α-TCT in 1% ethanol	●1% ethanol	●Significantly reduced the levels of ROS	●Significant strong protection of total cholesterol and free cholesterol.
[23]	●Alzheimer’s disease model using human neuroblastoma cells line [SH-SY5Y APP] overexpressing the human APP695 isoform	●Simultaneous treatment with 10 µM of α-TCT in 1% ethanol for 24 h	●1% ethanol	●A nonsignificant increase in the levels of Aβ indicating early onset of AD	●Direct activation of γ-secretase independent of gene expression
[23]	●Alzheimer’s disease model using human neuroblastoma cells line [SH-SY5Y wild-type]	●Simultaneous treatment with 10 µM of α-TCT in 1% ethanol for 24 h	●1% ethanol	●A significant increase in the levels of Aβ	●Due to direct increase in β-secretase activity independent of gene transcription of BACE1
[23]	●Alzheimer’s disease model using human neuroblastoma cells line [SH-SY5Y cells] stably expressing C99	●Simultaneous treatment with 10 µM of α-TCT in 1% ethanol for 24 h	●1% ethanol	●Significantly increased levels of Aβ	●Direct activation of γ-secretase independent of gene transcription of PSEN1, PSEN2, NCSTN, PSENEN and APH1A
[23]	●Alzheimer’s disease model using mouse neuroblastoma cell line (N2a)	●Simultaneous treatment with 10 µM of α-TCT in 1% ethanol for 24 h	●1% ethanol	●Significantly decreasing Aβ degradation	●Inhibiting insulin-degrading enzyme
[26]	●Glutamate neurotoxicity for 24 h using human astrocytes cell line (CRL-2020 cells) derived from glioblastoma with S100B protein	●5-min pre-treatment with 100, 200 and 300 ng/mL of TRF in absolute ethanol (TRF: 25% tocopherol and 75% tocotrienols)	●Absolute ethanol	●TRF could neither promptly (significantly) enhance cellular viability nor modulate the situation of oxidative stress since the level of the reduced glutathione was still low. However, 200 and 300 ng/mL could significantly attenuate lipid peroxidation through reducing the MDA level.	●Morphological cellular changes indicated a significantly reduction in the percentages of apoptotic and necrotic cells at higher concentrations.
[26]	●Glutamate neurotoxicity for 24 h using human astrocytes cell line (CRL-2020 cells) derived from glioblastoma with S100B protein	●30-min post-treatment with 100, 200 and 300 ng/mL of TRF in absolute ethanol (TRF: 25% tocopherol and 75% tocotrienols)	●Absolute ethanol	●TRF could neither promptly (significantly) enhance cellular viability nor modulate the situation of oxidative stress since the level of the reduced glutathione was still low. However, TRF could significantly attenuate lipid peroxidation through reducing the MDA level.	●Morphological cellular changes indicated a significant reduction in the percentages of apoptotic and necrotic cells at higher concentrations.
[30] *	●Alzheimer’s disease model with Aβ42 aggregates for 24 h using human neuroblastoma cell line (SH-SY5Y)	●Simultaneous treatment with 0.00003, 0.0003, 0.003% (v/v) TRF in 0.15% ethanol (TRF: 196 mg/g α-TCT, 24 mg/g β-TCT, 255 mg/g γ-TCT,75mg/gδ-TCT and 168 mg/g α-tocopherol)	●0.15% ethanol	●TRF could significantly enhance cellular viability	

AD: Alzheimer’s disease, TRF: tocotrienol-rich fraction, MDA: malondialdehyde, TCT: tocotrienol, Aβ: amyloid-β protein, ROS: reactive oxygen species. PSEN1: presenilin 1, PSEN2: presenilin 2, NCSTN: nicastrin, PSENEN: presenilin-enhancer 2, APH1A: anterior-pharynx-defective 1A, BACE1: Beta-secretase 1. * superscript: this study was a part of a preclinical animal study.

**Table 7 nutrients-12-00521-t007:** Characteristics of preclinical animal studies.

Reference	Human Modeled Disease, Study Design and Population	Intervention	Comparator	Outcomes	
				Primary	Secondary
[33]	●Nutritionally induced-cognitive dysfunction ●Young healthymale Crj:CD-1 (ICR) mice (3 weeks, *n* = 5) ●Long-term mono-level oral *ad libitum* intervention for 8 months.	●Palm oil (5 g/100 g NRD)	●100 g NRD	●Slight (nonsignificant) improvement in cognitive functions as evidenced by the non-significant reduced escape latency.	
[32]	●Diabetes-induced cognitive dysfunction ●Healthy male Wistar rats (age? *n* = 8). ●IP-injection of 45 mg/kg STZ (pH = 4.4, 0.1 M citrate buffer), while control was IP injected with citrate buffer vehicle. ●Long-term multilevel single oral daily intervention started from the 3rd day of STZ injection for 10 weeks.	●25, 50 or 100 mg/kg of TRF triturated with 5% tween 80 and dissolved in 5 mL/kg doubled distilled water. (TRF: Purity and composition was not stated)	●5% tween 80 in 5 mL/kg doubled distilled water	●Significant dose-dependent improvement in cognitive dysfunctions as evidenced by the deceased transfer latency (the time to reach the platform) and increased the time spent in the target quadrant (improved memory consolidation after learning). ●A significant dose-dependent improve in the cerebrocortical cholinergic activity, while the hippocampal cholinergic function was not significantly improved. ●A significant dose-dependent reversal of the cerebrocortical and hippocampal oxidative stress through attenuating lipid peroxidation and enhancing the activity of the antioxidant enzymes. ●A significant dose-dependent anti-inflammatory effect though reducing the cerebrocortical and hippocampal levels of TNF-α, IL-1β and p56 subunit of NFκβ. ●A significant dose-dependent antiapoptotic effect through reducing cerebrocortical and hippocampal levels of caspase-3.	
[32]	●Normal cognitive function ●Healthy male Wistar rats with (age? *n* = 8–10). ●Long-term mono-level single oral daily intervention for 10 weeks.	●100 mg/kg of TRF triturated with 5% tween 80 and dissolved in 5 mL/kg doubled distilled water. (TRF: Purity and composition was not stated)	●5% tween 80 in 5 mL/kg doubled distilled water.	●The cognitive performance was slightly (nonsignificant) increased as evidenced by the non-significantly reduced escape latency	
[36]	●Diabetes-induced cognitive dysfunction ●Healthy male Wistar rats, (Age? *n* = 5–8) were intracerebroventricularly injected with of 2 µL of 3 mg/kg STZ (pH = 4.4 and 0.1 M of citrate buffer) in two divided doses (on day 1 and day 3), while the comparator rats were intraventricular injected with 2 µL of citrate buffer (pH = 4.4, 0.1 M). Post-operative, rats were orally fed on milk and allowed to feed on NRD (*ad libitum*) for 4 days followed by feeding on NRD up to the end of the treatment. ●Short-term multilevel single oral daily intervention started by the 1st day of injecting STZ to be continued for 21 days.	●50 and 100 mg/kg α-TCT triturated with 5% tween and dissolved in double distilled water.	●5% tween and dissolved in double distilled water.	●A significant dose dependent improvement in cognitive functions as evidenced by the reduced escape latency. ●A significant dose-dependent reversal of neuro-oxidative stress through attenuating lipid peroxidation and enhancing of the activity of the antioxidant enzymes	
[38]	●Healthy cognitive function ●Healthy male Wister rats (age 3 months, *n* = 10) ● Long-term mono-level single oral daily intervention for 8 months.	●200 mg/kg TRF in 5 mL/kg of distilled water (TRF: Purity and composition were not stated)	●5 mL/kg distilled water	●Significantly enhanced cognitive functions as evidenced by the reduced escape latency ●Significantly reduced plasma DNA damage ●Significant reversal of serum oxidative stress through increasing the activity of antioxidant enzymes	
[34]	●Nutritionally induced cognitive dysfunction. ●Healthy male Sprague-Dawley rats (age? *n* = 10) ●Long-term mono-level of ad libitum oral feeding on intervention admixed with palm oil base vehicle. ●Rats exposed to the same intervention levels during gestation, 2 weeks during lactation, 8 weeks after weaning.	●100 mg/kg TRF suspended in 70 g/kg of palm oil base and admixed with 100 g NRD (TRF: Gold-Tri E ™70)	●70 g/kg of palm oil base admixed with 100 g NRD.	●Significant improvement in the cognitive functions of rats’ progeny ●Plasm and brain concentrations of tocotrienols indicated that α-TCT was the highest among the other isomers.	
[37]	●Chronic induced-stress condition ●Healthy male Sprague-Dawley rats (5 weeks, *n* = 9), which were stressed 5 h daily started from the 3rd week of intervention and continued for 21 days. ●Long-term mono-level single oral daily dose intervention for 5 weeks.	●200 mg/kg of TRF in normal saline (TRF: Tocomin^®^ Suprabio^TM^ 20%)	Normal saline		●Non-significant enhancement of the cellular proliferation and survival as well as expression of GAP-43 gene of granule cells in dentate gyrus
[37]	●Unstrained conditions ●Healthy male Sprague-Dawley rats (5 weeks, *n* = 9) ●Long-term mono-level single oral daily dose intervention for 5 weeks	●200 mg/kg of TRF in normal saline (TRF: Tocomin^®^ Suprabio^TM^ 20% but compostion was not stated)	Normal saline		●No significant alteration in the cellular proliferation and survival as well as expression of GAP-43 gene of granule cells in dentate gyrus
[39]	●Healthy cognitive function ●Healthy young male Wister rats (3 months, *n* = 9) ●Long-term mono-level oral single daily intervention for 3 months	●200 mg/kg TRF in 5 mL/kg of olive oil (TRF = 149.2 mg/g α-tocopherol, 164.7 mg/g α-TCT, 48.8 mg/g β-TCT,213.2 mg/g γ-TCT and 171 mg/g δ-TCT).	5 mL/kg of olive oil	●No significant alteration in the cognitive functions as evidenced by the non-significant difference in escape latency. ●no significant alteration in plasma lipid peroxidation and the plasma activity of antioxidant enzymes ● A slight (nonsignificant) reduction in plasma DNA damage	
[39]	●cognitive dysfunction ●elderly male Wister rats (21 months, *n* = 9) ●Long-term mono-level oral single daily intervention for 3 months	●200 mg/kg TRF in 5 mL/kg of olive oil (TRF = 149.2 mg/g α-tocopherol, 164.7 mg/g α-TCT, 48.8 mg/g β-TCT, 213.2 mg/g γ-TCT and 171 mg/g δ-TCT).	5 mL/kg of olive oil	●Significant Improved cognitive functions as evidenced by the significant reduction in escape latency. ●Reversal of the plasma oxidative stress through a significant attenuating lipid peroxidation and enhancing the activity of oxidative enzymes ● Significant attenuation of plasma DNA damage	
[30]	●Transgenic Alzheimer’s disease ●Heterozygous AβPP/PS1 double transgenic male mice (human chimeric amyloid expressing precursor protein and a mutant human presenilin 1 with deletion at exon 9), (5 months) ●Long-term mono-level single oral daily intervention for 10 months.	●60 mg/kg of TRF in 5 mL/kg 12mg/mL vitamin-E-striped palm oil (*n* = 11) (TRF = 196.0 mg/g α-TCT, 24 mg/g β-TCT, 255 mg/g γ-TCT, 75 mg/gδ-TCT and 168 mg/g α-tocopherol)	●5 mL/kg of 12 mg/mL of vitamin-E-striped palm oil (*n* = 10)	●Slight (nonsignificant) enhancement of the recognition functions as evidenced by the nonsignificant increase in the recognition index, but the location preference was equivalent as evidenced by the equal spent time to explore the identical objects. ●A non-significant change in the levels of soluble and insoluble cortical or hippocampal Aβ isoforms (Aβ _40_, Aβ _42_ and Aβ oligomer).	●A slight (nonsignificant) reduction in the hippocampal Aβ deposition, but significant reduction in the cortical Aβ deposition ●A significant reduction in cortical and hippocampal *A*β plaques
[35]	●Transgenic Alzheimer’s disease ●Heterozygous AβPP/PS1 double transgenic male mice (human chimeric amyloid expressing precursor protein and a mutant human presenilin 1 with deletion at exon 9), (9 months, *n* = 4) ●Long-term mono-level single oral daily intervention for 6 months.	●200 mg/kg TRF in 12 mg/mL vitamin E striped palm oil(TRF = f 24% α-tocopherol, 27% α-TCT,4% β-TCT, 32% γ-TCT,and 14% δ-TCT)	12 mg/mL of vitamin E striped palm oil		Significant upregulation of genes responsible for neuroprotective effects such as Slc24a2, exo1 and Enox1 ●Significant downregulation of genes responsible for the pathology of AD such as Pla2g4a and Tfap2b
[31]	●Transgenic Alzheimer’s disease ●Heterozygous AβPP/PS1 double transgenic male mice (human chimeric amyloid expressing precursor protein and a mutant human presenilin 1 with deletion at exon 9), (5 months, *n* = 9). ●Long-term mono-level single oral daily intervention for 10 months.	●60 mg/kg TRF in 12 mg/mL of vitamin-E-striped palm oil(TRF = 23.40% α-tocopherol (23.40%),27.30% α-TCT; 3.34% β-TCT, 35.51% γ-TCT and 10.45% δ-TCT.)	●5 mL/kg of vitamin-E-stripped palm oil	●Slight (nonsignificant) enhancement of cognitive functions as evidenced by non-significantly reduced escape latency.	●TRF could alter 90 putative metabolites involved in several metabolic AD-related pathways.

AD: Alzheimer’s disease; TRF: Tocotrienol-rich fraction; MDA: malondialdehyde, TCT: tocotrienol, Aβ: amyloid-β protein, NRD: normal rodent diet, *n*: number of animals per group of either intervention or comparator, STZ: streptozotocin, ROS: reactive oxygen species, SOD: superoxide dismutase, CAT: Catalase, GPx: glutathione peroxidase; GSH: reduced glutathione, ?: not stated in the study, IP: intraperitoneal, Slc24a2: Solute carrier family 24 [sodium/potassium/calcium exchanger], exo1: Exonuclease 1, Enox1: Ecto-NOX disulfide-thiol exchanger 1, Pla2g4a: Phospholipase A2, group IVA [cytosolic, calcium-dependent], Tfap2b: Transcription factor AP-2 beta.

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
