# Peer review of "Safety and Neuroprotective Efficacy of Palm Oil and Tocotrienol-Rich Fraction from Palm Oil: A Systematic Review"

_nutrients, 2020, doi:10.3390/nu12020521_

Round 1

Reviewer 1 Report

The paper provides a thorough review of the effects of palm oil on neurons and, consequently, cognition. The conclusions, which are supported by the literature included in the review, is that the effects are modestly positive with very little downside.

I think the review achieves what it set out to do. The criteria for determining whether a study employed methods that may lead to biased conclusions were clear, and the 18 included studies were likely of high quality, although I am not familiar with them directly.

I do feel that the review could be substantially improved with two sets of revisions.

1) While cognitive function is a central idea in the review, it is only mentioned once (in a parenthetical) that cognitive ability corresponds to how long it takes a mouse or rat to escape a maze. It would be helpful to report the cognitive assessment used in each applicable study. Even if they are all very similar, it would be helpful to know in more detail what this task involves.

2) The effects of palm oil (and active elements) are only qualitatively described at all levels of analysis. Including effect sizes would be extremely valuable. An effect can be statistically significant and very small. Part of the utility of a review such as this should be to give readers context: not just that effects exist but how strong are they?

The combination of these two things will greatly improve the utility of the document, because readers will be more able to appreciate what aspects of cognition are being improved and the magnitude of these improvements.

In addition: The paper has a few typos and minor English errors. Nothing major--it was a very easy read in that respect.

Reviewer 2 Report

This review article is based on PRISMA specifications to search literature data and write manuscript. Also the authors also performed the assessment of risk of bias in vitro studies and in animal experiments via OHAT and SYRCLE’s tools. This review article has been well designed and analyzed. I have some minor comments described below:

Please add the comparison of TCT and TRF. TCT is mostly used in vitro studies, but TRF is mostly used in animal experiments. Why? Whose purity of TRF? Please compile all literatures. Please rewrite line 522 – 553 page 33. This discussion is very confused. I think that the difference is mainly to the blood brain barrier in natal and adult, normal and disease.

Reviewer 3 Report

This paper is reviewed on the effect of palm oil and TRF on the cognitive performance of healthy animals. This review will be accepted in the publication of Nutrients.

The authors systematically summarized the effect o palm oil and TRF. On the other hand, palm oil is composed of palmitic acid, oleic acid, and so on. The authors should mention the individual effects on these organic acids.  In addition, if the authors mention on the effects of organic acids on the chemical systems, e.g., membrane system, spectroscopy, chemical researchers have an interest in scientific discussion from view point of molecular level. Tables should be used as landscape display to read them.
